# Immune profiling of mpox survivors reveals divergent durability of antibody and T cell responses

Yanqun Wang [1,2,9] ✉, Ruoxi Cai[1,9], Airu Zhu[1,9], Jiantao Chen[1,9], Canjie Chen[1,9], Lijuan Zhou[3,4,9], Xindan Xing[1,9], Qier Zhong[1,2], Peilan Wei[1,5], Xinxin Li[1], Zhaoyong Zhang[1], Yuanyuan Zhang[1], Lei Chen[1], Jingjing Gao[1], Suxiang Li[6], Xinyi Xiong[7], Bin Qu[7], Shuxiang Huang[6], Zhiwei Lin[6], Haoshi Bai[1], Qingtao Hu [1], Jingxian Zhao [1,5], Yongxia Shi[6], Yang Yang [8] ✉, Pengzhe Qin[3,4] ✉, Lu Zhang [6] ✉ & Jincun Zhao [1,2,5,7] ✉

Despite the global spread of mpox virus (MPXV), the durability and breadth of infection-induced immunity remain incompletely defined. Here, we comprehensively characterize MPXV-specific antibody and T cell responses up to 18 months after natural infection in male individuals. Neutralizing antibodies exhibit typical acute viral kinetics, with titers peaking early and declining over time, from a mean of 328 at 12 months to 180 at 18 months post-infection. Neutralization analyses against MPXV clade Ib, clade IIb and VACV WR strains demonstrate pronounced cross-neutralization among orthopoxviruses, but with lineage-specific reductions in neutralization, indicating incomplete cross-reactivity across different lineages. In parallel, MPXV-specific CD4$^+$ and Tfh cell responses remain robust and polyfunctional throughout follow-up, and CD8$^+$ T cells maintain sustained responses characterized by cytokine production, together supporting durable cellular immunity. These findings offer critical insights into the durability and breadth of post-infection immunity, with implications for reinfection risk and orthopoxvirus vaccine strategies.

Since 2022, mpox virus (MPXV), a zoonotic orthopoxvirus closely related to variola virus (VARV), has triggered a global outbreak of unprecedented scale and geographic reach. Initial clusters in non-endemic regions were concentrated among men who have sex with men (MSM), but subsequent epidemiology has demonstrated wider involvement of women and children in both endemic and non-endemic settings[1–4]. In contrast to the sporadic, geographically restricted outbreaks of previous decades, the current epidemic is characterized by sustained human-to-human transmission across multiple continents[5–7]. This expanded epidemiology underscores the

[1]State Key Laboratory of Respiratory Disease, National Clinical Research Center for Respiratory Disease, Guangzhou Institute of Respiratory Health, the First Affiliated Hospital of Guangzhou Medical University, Guangzhou, China. [2]GMU-GIBH Joint School of Life Sciences, Guangzhou Medical University, Guangzhou, China. [3]Guangzhou Center for Disease Control and Prevention (Guangzhou Health Supervision Institute), Guangzhou, China. [4]Institute of Public Health, Guangzhou Medical University & Guangzhou Center for Disease Control and Prevention (Guangzhou Health Supervision institute), Guangzhou, China. [5]Guangzhou National Laboratory, Guangzhou, China. [6]Health and Quarantine Laboratory, State Key Laboratory of Respiratory Disease of Guangzhou Customs District Technology Center, Guangzhou, China. [7]Shanghai Institute for Advanced Immunochemical Studies, School of Life Science and Technology, ShanghaiTech University, Shanghai, China. [8]National Clinical Research Center for Infectious Disease, Shenzhen Third People's Hospital, Second Hospital Affiliated to Southern University of Science and Technology, Shenzhen, China. [9]These authors contributed equally: Yanqun Wang, Ruoxi Cai, Airu Zhu, Jiantao Chen, Canjie Chen, Lijuan Zhou, Xindan Xing. ✉e-mail: wangyanqun@gird.cn; young@mail.sustech.edu.cn; petgyy@gmail.com; zhanglu0307@163.com; zhaojincun@gird.cn

urgent need for strengthened surveillance and immunological investigations across diverse populations. Although most infections are self-limiting, the clinical burden, including painful skin lesions, secondary bacterial infections, and serious complications, poses significant public health challenges[8,9].

Notably, individuals younger than 44 years of age (born after 1981 in China, when routine smallpox vaccination ceased; the year of discontinuation varies by country) generally lack pre-existing orthopoxvirus immunity. The resulting absence of population-level protection raises concerns about increased susceptibility to MPXV infection and the potential for sustained transmission. These concerns highlight the urgent need to define the durability and breadth of infection-induced immunity, while recent reports of reinfection further underscore the importance of temporal immune profiling[10,11].

Recent studies have begun to delineate the humoral immune response to MPXV infection, showing rapid induction of binding and neutralizing antibodies during the acute and early convalescent phases[12–14]. Yet the durability and functional quality of these responses, particularly beyond the first year post-infection, remain poorly defined. Most reports are limited to 3–12 months of follow-up[12,15,16], leaving critical gaps in our understanding of long-term immune memory. Early data suggest that anti-MPXV antibody titers wane and T cell memory responses are variable in magnitude and quality[17,18], but studies with >12 months of follow-up or high-resolution profiling are largely absent. The lack of comprehensive temporal immunoprofiling, including assessments of neutralization breadth, antibody durability, and T cell functionality, continues to hinder evaluation of reinfection risk, refinement of vaccination strategies, and identification of correlates of durable protection.

To address the critical gap in understanding long-term orthopoxvirus immunity, we analyzed immune responses in individuals recovering from MPXV infection over an 18-month period. Through serial immunoprofiling of samples collected at multiple time points from different participants, we characterized both humoral and cellular immunity, revealing a classical acute-phase antibody trajectory with an early peak followed by a gradual decline. Cross-neutralization assays against divergent orthopoxviruses, including MPXV clades Ib and IIb and vaccinia virus (VACV), demonstrated broad cross-reactivity but lineage-specific reductions in potency. These findings provide key insights into the durability and breadth of infection-induced immunity, with direct implications for reinfection risk and evaluation of orthopoxvirus vaccine strategies.

## Results

### Baseline characteristics of the study cohorts
Between August 2022 and October 2024, we enrolled a total of 40 men with PCR-confirmed MPXV infection, all identified as belonging to clade IIb, the lineage circulating in China during this period. In detail, 17 male participants were recruited during the acute phase at Shenzhen Third People's Hospital, yielding 26 plasma samples, 13 collected at 10 days post-onset (10 d.p.o) and 13 at 20 d.p.o., with several individuals contributing samples at both timepoints. An additional 23 male convalescent individuals were enrolled by the Guangzhou Center for Disease Control and Prevention, providing 48 clinical samples collected temporally at 6, 9, 12, and 18 months post-onset to assess the durability of humoral responses. In total, samples were obtained across six time points (10 and 20 days; 6, 9, 12, and 18 months post-onset) depending on availability. For comparison, we included 30 age-matched healthy controls with no history of orthopoxvirus exposure and 30 individuals with documented smallpox vaccination (Table 1).

Across the infected cohort, 37.5% (15/40) were living with HIV. The median age was 30 years, and all participants self-identified as MSM. None had received smallpox vaccination within the past decade (Supplementary Table 1). All mpox cases were symptomatic but classified as moderate by attending clinicians. Disease severity was determined according to established clinical criteria, including systemic symptom persistence (e.g., sustained fever), extent of cutaneous lesions (localized versus widespread or necrotic), clinical course (self-limited versus secondary bacterial infection), and organ involvement. No severe cases or fatalities were reported. This well-characterized cohort provides a valuable opportunity to evaluate the kinetics, durability, and breadth of post-infection immunity following moderate MPXV clade IIb infection, including both HIV-positive and HIV-negative individuals.

### Antigen-specific heterogeneity and temporal decline in MPXV binding antibody responses
IgG responses against key MPXV antigens (A35, A29, B6, E8, H3, and M1) were assessed by enzyme-linked immunosorbent assay (ELISA) at multiple time points over an 18-month follow-up. Overall, a progressive decline in antigen-specific IgG titers was observed in convalescent individuals. Seroconversion rates for A35, A29, B6, and E8 reached 100% by day 20 post-onset of symptoms, followed by gradual waning over time, consistent with the contraction phase of acute antibody responses. In contrast, responses to H3 and M1 were less frequent, underscoring heterogeneity in the humoral response to distinct MPXV antigens (Fig. 1a).

During the recovery phase (9–18 months POS), IgG detection rates remained variable: 50% (range 30–70%) for A35, 8.7% (range 6.25–20%) for A29, 58.7% (range 50–70%) for B6, 69.5% (range 60–90%) for E8, 56.5% (range 40–80%) for H3, and 23.9% (range 18.75–30%) for M1 (Fig. 1a). Temporal analysis of IgG titers over 18 months confirmed gradual waning after peak responses (Fig. 1b). Similar kinetics were observed for VACV antigens A33, A27, D8, and H3 (Supplementary Figs. 1 and 2), and IgG binding to MPXV surface antigens significantly correlated with responses to their VACV orthologs (Supplementary Fig. 3). As expected, healthy controls showed no reactivity, whereas historically smallpox-vaccinated individuals exhibited low but detectable IgG titers against MPXV membrane proteins. Together, these data reveal a clear hierarchy of antigenic immunodominance among MPXV membrane proteins and highlight the progressive but heterogeneous decline of binding antibody responses over time.

### Temporal patterns of neutralizing antibody responses in mpox convalescents
We established a focus reduction neutralization test (FRNT) to quantify neutralizing antibody responses against both MPXV and VACV across four cohorts: acute MPXV infection, MPXV convalescents, historically VACV-immunized individuals, and unvaccinated controls. As expected, neutralizing activity was negligible in unvaccinated controls and weak in historically vaccinated individuals. By contrast, MPXV convalescents mounted robust cross-neutralizing responses to both MPXV and VACV, underscoring the strong induction of orthopoxvirus immunity after natural infection (Fig. 2a, b). Titers rose sharply during acute infection and subsequently waned, with mean MPXV-specific neutralizing titers declining from 328 at 12 months to 180 at 18 months post-infection, yet remaining markedly above the 67 observed in historically vaccinated individuals. Temporal assessment confirmed a consistent decline in neutralizing titers against MPXV, with mean values decreasing between 12 and 18 months post-infection. (Fig. 2e). A similar waning trajectory was observed for VACV-specific neutralization (Fig. 2d, f). We further compared neutralizing responses by HIV status. Convalescent individuals living with HIV exhibited neutralizing titers against MPXV clade IIb that were comparable to those of HIV-negative participants. Similarly, responses to other orthopoxviruses, including VACV, did not differ significantly between the two groups (Supplementary Fig. 4).

To further assess responses against distinct viral forms, we measured extracellular enveloped virion (EEV)-specific neutralization in

**Table 1 | The characteristics information of participants in this study**

|  |  | MPXV-cases (acute, 10–20D) | MPXV-recovery (long-term, 6–18 M) | Historically, VACV-vaccinated controls (non-infected) | Unvaccinated controls (non-infected) |
|---|---|---|---|---|---|
| No. of participants |  | 17 | 23 | 30 | 30 |
| No. of samples |  | 26 | 48 | 30 | 30 |
| Age, yrs (IQR) |  | 32 (25–51) | 29.8 (19–43) | 71 (59–86) | 32.2 (24–41) |
| Sex, n (%) | Female | 0 (0%) | 0 (0%) | 8 (27%) | 23 (77%) |
|  | Male | 17 (100%) | 23 (100%) | 22 (73%) | 7 (23%) |
| Severity, n (%) | Moderate | 17 (100%) | 23 (100%) | - | - |
| MPXV genotypes |  | Clade IIb | Clade IIb | NA | NA |
| MPI, samples (%) | 10 D | 13 (50%) | - | - | - |
|  | 20 D | 13 (50%) | - | - | - |
|  | 6 M | - | 2 (4%) | - | - |
|  | 9 M | - | 10 (21%) | - | - |
|  | 12 M | - | 16 (33%) | - | - |
|  | 15 M | - | 10 (21%) | - | - |
|  | 18 M | - | 10 (21%) | - | - |

D day, M month, NA not available, MPI month post-infection.

convalescent plasma. Neutralizing activity against EEVs was consistently detectable throughout follow-up and exhibited similar decay kinetics, albeit at lower magnitude than assays using mixed MPXV virions, reflecting methodological differences between assay formats (Supplementary Figs. 5 and 6). Importantly, neutralizing antibodies persisted in all convalescent individuals at 18 months, even when binding antibody responses to individual MPXV antigens had waned (Fig. 1). Together, these findings indicate that neutralization assays serve as a sensitive measure of long-term functional antibody activity and complement binding assays in assessing MPXV immunity.

### Lineage-specific reductions in neutralization reveal antigenic divergence among orthopoxviruses

The global spread of MPXV and the reliance on VACV-based vaccines highlight the need to define the breadth of cross-neutralizing immunity across orthopoxvirus lineages. To this end, we performed live-virus $FRNT_{50}$ assays using MPXV clade IIb, MPXV clade Ib, and VACV (WR strain) (Fig. 3a, b). We analyzed 26 plasma samples from 17 individuals during acute infection (10–20 days post-onset) and 48 samples from 23 convalescents collected 6–18 months post-infection.

During acute infection, plasma from MPXV clade IIb-infected individuals exhibited broad but heterogeneous cross-neutralizing activity. Neutralization titers against clades IIb and Ib were nearly identical (mean 132 vs. 130), whereas titers against VACV WR were lower (mean 68), indicating antigenic divergence (Fig. 3a). At later timepoints, cross-neutralization efficiency declined further, with convalescent plasma showing a 1.6-fold reduction against clade Ib and a threefold reduction against VACV relative to homologous MPXV IIb (Fig. 3b). Similarly, plasma from VACV-vaccinated individuals displayed diminished activity against both MPXV clades, with reductions of 1.5- and 1.7-fold, respectively (Fig. 3c).

These results demonstrate that prior infection or vaccination elicits cross-reactive but incomplete immunity across divergent orthopoxvirus lineages. The observed reductions in neutralization potency likely reflect antigenic divergence among major orthopoxvirus antigens (Supplementary Fig. 7), consistent with reports of low MPXV-neutralizing titers following MVA-BN vaccination in healthy individuals. Comparative genomic analyses of MPXV clades I and II and VACV further support this divergence (Fig. 3d). Together, these findings reveal broad yet variable cross-neutralization and underscore the need for strategies to enhance protection across orthopoxvirus lineages.

### Principal component analysis reveals antigenic correlates of neutralization in MPXV convalescents

Principal component analysis (PCA) was performed to integrate antibody binding and neutralizing titers against MPXV and VACV, providing a dimensionality-reduction framework to visualize immune profiles across exposure groups. MPXV convalescents showed uniform clustering, while individuals with active infection displayed partial separation but with some overlap (Fig. 4a, b). Compared with controls, both infected and convalescent individuals diverged from unvaccinated and historically vaccinated participants, though overlap remained, indicating that PCA captures overall trends rather than strict categorical boundaries.

To identify immunological variables associated with these patterns, a PCA biplot was generated (Fig. 4c), which suggested that antibody responses directed against E8, A29, A35, and M1, together with MPXV-specific $FRNT_{50}$ values, contributed most strongly to the observed clustering of active and convalescent groups. Further analysis of MPXV convalescents (Fig. 4d) revealed strong correlations between antigen-specific IgG binding and neutralizing activity, with the most robust associations observed for A35, followed by E8 and B6. These results suggest that multiple MPXV antigens correlate with neutralization, although validation with mono-specific sera will be required to establish causality.

### Sustained, polyfunctional MPXV-specific T cell responses persist up to 18 months after infection

To assess MPXV-specific T-cell responses during convalescence, post-infection PBMCs were stimulated with heat-inactivated MPXV (clade IIb) infected Vero 81 cell supernatant or control non-infected Vero 81 cell supernatant (Fig. 5 and Supplementary Fig. 8). T-cell activation was quantified by intracellular cytokine staining for interferon-γ (IFN-γ) and tumour necrosis factor (TNF). Representative flow cytometry plots for MPXV antigen stimulations and negative controls are provided in Supplementary Fig. 9. Analysis of PBMCs collected at 9, 12, 15 and 18 months post-symptom onset revealed durable MPXV-specific responses in both $CD4^+$ and $CD8^+$ T cells. By 18 months, all convalescents retained detectable $CD4^+$ responses and 80% maintained $CD8^+$ responses. Compared with uninfected or VACV-vaccinated individuals, MPXV convalescents exhibited significantly higher frequencies of virus-specific T cells ($p < 0.01$; Fig. 5a, b). In addition to conventional helper T cell responses, circulating T follicular helper (cTfh) cells displayed sustained activation, defined by ICOS and CD40L

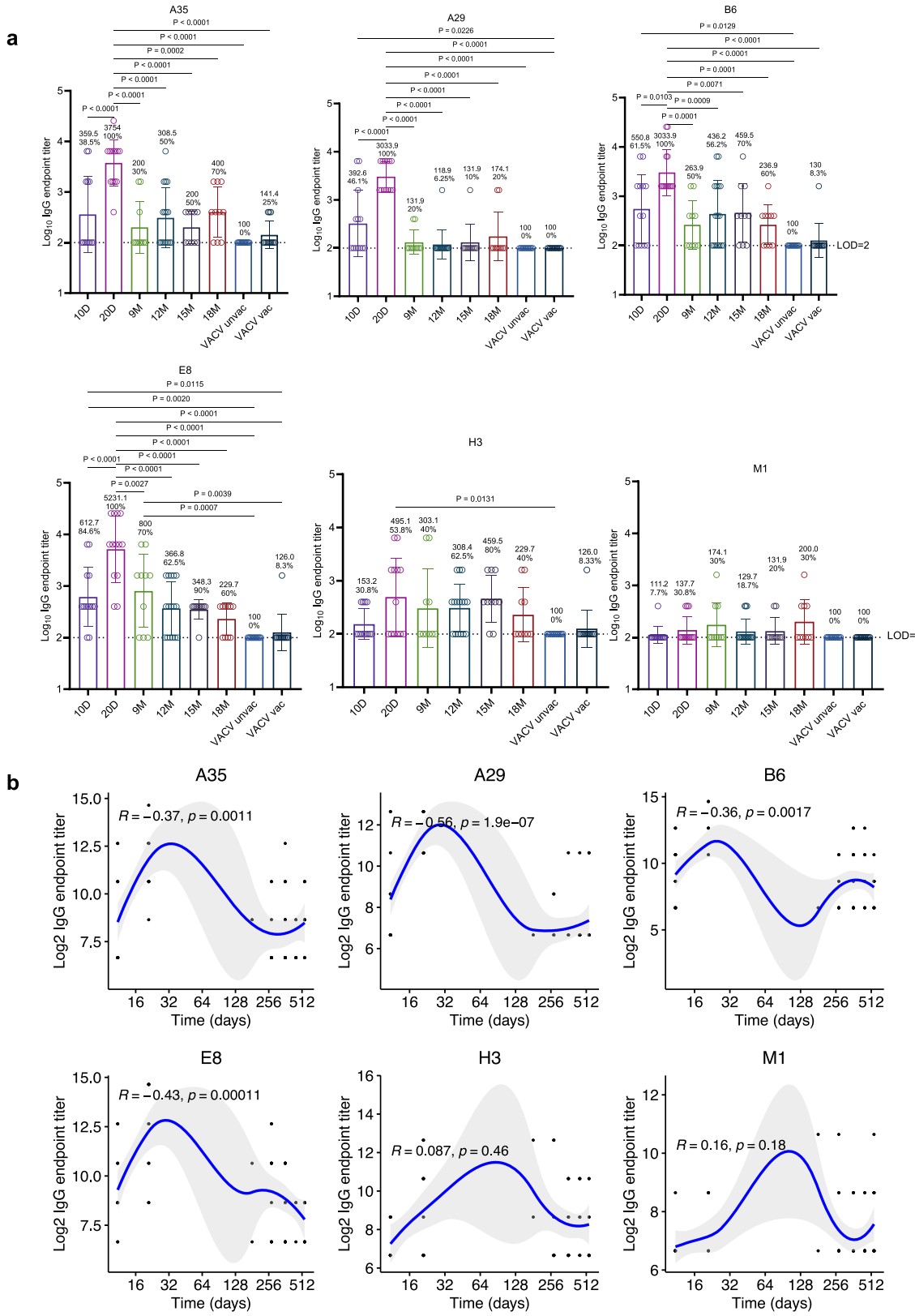

co-expression, throughout convalescence (Supplementary Fig. 10). MPXV-specific cTfh responses remained detectable in all convalescent individuals at 18 months (100%; Fig. 5c), implicating sustained Tfh-B cell interactions in the maintenance of long-term antibody-mediated immunity. We further analyzed the relationship between cTfh cell frequencies and both MPXV-specific neutralizing and binding antibody titers, but no significant correlations were observed (Supplementary

Fig. 11). In addition, cTfh frequencies did not differ significantly between HIV-positive and HIV-negative convalescents (Supplementary Fig. 12), although larger cohorts will be required to confirm these findings.

To further evaluate the functional competence of MPXV-specific T cells, we examined their cytokine production profiles. Both CD4+ and CD8+ T cells were detectable following infection. MPXV-specific CD4+

**Fig. 1 | Durability of plasma IgG responses to MPXV antigens after recovery.**
**a** Plasma IgG binding responses against major MPXV surface antigens (A35, A29, B6, E8, H3, and M1) were detected by ELISA. Endpoint titers of antigen-specific IgG antibodies were determined for each time point. Four participant groups were included in this study: (1) acute-phase mpox patients, from whom blood samples were collected at 10 days ($n = 13$) and 20 days ($n = 13$) after symptom onset; (2) mpox convalescents who were recruited and followed for up to 18 months post-onset (9 M, $n = 10$; 12 M, $n = 16$; 15 M, $n = 10$; 18 M, $n = 10$); (3) a VACV-unvaccinated cohort ($n = 30$); and (4) a VACV-vaccinated cohort ($n = 30$). Mean endpoint titers and seropositivity rates are indicated above each column. All assays were conducted with three technical replicates per sample. Statistical significance was evaluated using one-way ANOVA followed by Tukey's multiple comparisons test. All analyses were conducted two-sided. Error bars represent mean ± SD. Statistical significance is denoted as: ns ($P > 0.05$), *$P < 0.05$, **$P < 0.01$, ***$P < 0.001$, ****$P < 0.0001$. Only comparisons with $p \leq 0.05$ are shown. **b** The kinetics of MPXV-specific IgG endpoint titers over an 18-month period post-recovery were visualized using scatter plots with locally estimated scatterplot smoothing (LOESS). Blue lines represent the fitted curve obtained using the LOESS curve fitting polynomial regression, and the gray band areas represent 95% confidence intervals. Pearson correlation coefficients and two-sided corresponding $p$-values were computed and displayed on the plots.

T cells exhibited a markedly polyfunctional phenotype, with a significantly higher proportion of cells co-producing TNF or IL-2 together with IFN-γ compared to responses induced by prior VACV vaccination. This enrichment suggests enhanced capacity for rapid and efficient recall upon MPXV re-exposure. MPXV-specific CD8+ T cells generated sustained cytokine-producing responses, though with lower polyfunctionality than CD4+ T cells. (Supplementary Fig. 13). Together, these findings indicate that both CD4+ and CD8+ T cells contribute to the cellular immune landscape following MPXV infection.

In parallel, we observed a pronounced increase in CD4+ effector memory T cells (TEM; CCR7−CD45RA−) relative to VACV-vaccinated individuals, consistent with the establishment of a robust effector memory compartment following natural infection (Supplementary Fig. 14). The persistence of TEM cells for up to 15 months post-symptom onset suggests durable effector memory poised for rapid recall upon re-exposure. Collectively, these findings demonstrate that natural MPXV infection induces durable virus-specific T cell memory, encompassing both CD4+ and CD8+ compartments. This coordinated cellular immunity is likely to support long-term protection and complement waning humoral responses.

## Discussion

Although MPXV continues to spread globally, the durability of immune responses following natural infection, particularly regarding long-term protection against reinfection, remains poorly defined[12,19]. Comprehensive temporal assessments of both humoral and cellular immunity in MPXV convalescents over extended periods remain limited. In this prospective cohort study, we profiled MPXV-specific immune responses across an 18-month period, offering an integrated characterization of antibody and T cell kinetics following infection.

Our findings demonstrate that the trajectory of neutralizing antibody responses following MPXV infection mirrors that of typical acute viral infections, with antibody titers peaking early and subsequently declining over time. Notably, although neutralizing activity decreased substantially during convalescence, persistent antibody responses were still detectable at 18 months, suggesting partial potential protection against circulating MPXV strains. In parallel, we observed durable virus-specific T cell immunity persisting throughout the study period, particularly within the CD4+ and cTfh compartments, which are likely to contribute to sustained B cell help and long-term humoral memory[20,21]. These findings are consistent with recent reports highlighting the importance of T cell immunity in mpox. Adamo et al. showed that smallpox vaccination generates long-lived cross-reactive CD8+ memory T cells, while mpox convalescents develop polyfunctional CD8+ effector responses linked to milder disease[22]. Similarly, Chen et al. demonstrated that MPXV infection induces CD8+ memory T cells with stronger effector and migratory capacity than those elicited by MVA-BN vaccination[23]. Together with our data showing robust and polyfunctional MPXV-specific CD4+ and Tfh responses up to 18 months post-infection, these observations suggest that coordinated CD4+, Tfh, and CD8+ T cell immunity underpins durable protection against mpox.

Furthermore, the observation of lineage-specific reductions in cross-neutralizing activity underscores antigenic divergence among orthopoxviruses. Although these differences suggest that prior infection or vaccinia-based vaccination may not confer uniform cross-protection across divergent lineages, the clinical significance of reduced titers remains uncertain due to the lack of established correlates of protection in humans[19,24]. Collectively, these findings provide critical insights into both the durability and limitations of post-infection immunity, informing future strategies for booster vaccination and immunological risk assessment in at-risk populations.

In interpreting our results, host factors such as sex, HIV status, age and comorbidities should be considered. All MPXV-infected participants in this study were male, limiting the assessment of sex-related differences in infection-induced immunity. Although no sex-based differences in neutralizing activity were observed among VACV-vaccinated controls, validation in female mpox patients will be necessary to establish broader generalizability. With respect to HIV status, stratified analyses showed largely comparable immune responses, with HIV-positive participants showing similar neutralizing titers against MPXV clade IIb as HIV-negative individuals. The two cohorts (acute and convalescent) were also comparable in median age and disease severity, reducing the likelihood of residual confounding. Nevertheless, larger and more diverse cohorts will be required to further elucidate the influence of demographic and clinical factors on the durability and breadth of MPXV-induced immune responses.

Our results have several implications for vaccine and public health strategies. The waning of neutralizing antibody titers over time suggests that protection conferred by natural infection may diminish, highlighting the need for continued monitoring and possibly booster immunizations in high-risk populations. Importantly, our data indicate that clade IIb MPXV infection induces cross-neutralizing responses against other orthopoxviruses, albeit with reduced potency against some lineages. While current vaccinia-based vaccines have historically provided broad protection, the observed antigenic differences underscore the need for further evaluation of vaccine design and effectiveness against divergent orthopoxvirus lineages. Moreover, the sustained Tfh, CD4+ and CD8+ T cell responses highlight the importance of vaccines that not only induce durable humoral immunity but also robust cellular immunity.

This study has several limitations. First, the overall cohort size was modest, and participants were divided into two groups with either short-term ($n = 17$, 10–20 days post-onset) or long-term follow-up ($n = 23$, 6–18 months post-onset), without complete longitudinal datasets across all timepoints for any single individual. Consequently, comparisons between acute and convalescent phases rely on group-level rather than fully paired analyses. Second, the number of samples available at certain late timepoints was particularly limited, with only two samples at 6 months and nine samples at 18 months, none of which were matched to baseline acute-phase samples. These constraints reduce the statistical power of our analyses and highlight the need for larger cohorts with complete serial sampling to more precisely define the kinetics of long-term immunity. Third, all participants experienced mild disease, limiting the generalizability of our findings to individuals with severe mpox. Future studies incorporating more diverse cohorts and extended follow-up will be critical to address these limitations and strengthen the conclusions drawn here.

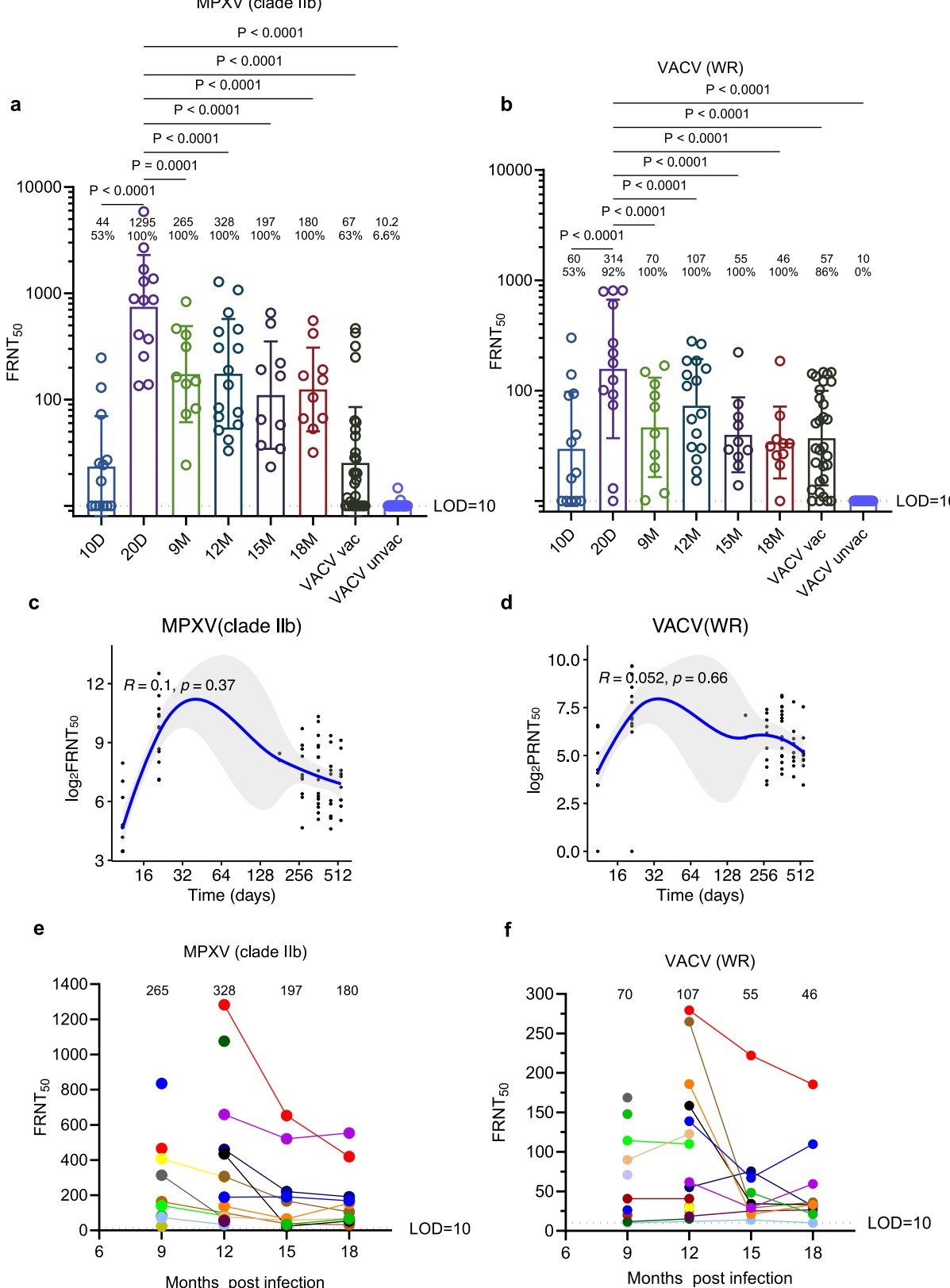

## Methods

### Viruses and cell lines

MPXV isolates representing clades Ib and IIb were obtained from lesion swabs of PCR-confirmed patients and propagated in Vero 81 cells under biosafety level 3 (BSL-3) containment at the Guangzhou Customs District Technology Center. The MPXV clade IIb strain was isolated in June 2023, and the clade Ib strain in February 2025, from local residents in Guangdong, China, who had recently been in close contact with international travelers arriving from mpox-endemic regions. Viral genome sequencing confirmed the lineage identities, and the sequences have been deposited in GenBank under accession numbers PX488348 (clade IIb) and PX488349 (clade Ib). Collection of

**Fig. 2 | Temporal kinetics of neutralizing antibody responses up to 18 months after MPXV infection. a, b** Neutralizing antibody responses against orthopoxviruses in MPXV-infected individuals (10–20 days POS) and convalescents (6–18 months POS). Neutralization titers against MPXV clade IIb (**a**) and VACV WR (**b**) strains were determined by focus reduction neutralization test (FRNT) in the presence of 10% guinea pig serum as a source of complement. Dotted lines indicate the limit of detection (LOD = 10). Neutralizing titer of VACV-unvaccinated control samples is undetectable and approaching baseline. Four participant groups were included in this study: (1) acute-phase mpox patients, from whom blood samples were collected at 10 days ($n = 13$) and 20 days ($n = 13$) after symptom onset; (2) mpox convalescents who were recruited and followed for up to 18 months post-onset (9 M, $n = 10$; 12 M, $n = 16$; 15 M, $n = 10$; 18 M, $n = 10$); (3) a VACV-unvaccinated cohort ($n = 30$); and (4) a VACV-vaccinated cohort ($n = 30$). All assays were conducted with three technical replicates per sample. Mean neutralizing titers and the corresponding seropositivity rates are indicated above each column. Statistical significance was assessed using one-way ANOVA followed by Tukey's multiple comparisons test. All analyses were conducted as two-sided. Error bars represent mean ± SD, P-values are displayed as ns for $P > 0.05$, *$P < 0.05$, **$P < 0.01$ and ***$P < 0.001$. Only comparisons with $p \leq 0.05$ are shown. Temporal trajectories of neutralizing antibody titers against MPXV clade IIb (**c**) and VACV WR (**d**) during the 18-month follow-up. Neutralizing antibody kinetics after recovery were visualized using scatter plots with locally estimated scatterplot smoothing (LOESS). Blue lines represent the fitted curve obtained using the LOESS curve fitting polynomial regression, and the gray band areas represent 95% confidence intervals. Pearson correlation coefficients and corresponding two-sided p-values were computed and displayed on the plots. Dynamic changes in neutralizing antibody responses against MPXV (**e**) and VACV (**f**) in individual MPXV convalescents, with all available data points shown. Each column presents the mean neutralizing titers.

lesion swabs for viral isolation was conducted under approval from the Institutional Review Board of Guangzhou CDC (GZCDC-ECHR-2023P0059). Written informed consent was obtained from all participants prior to sample collection and laboratory testing. Working stocks of MPXV were subsequently expanded and titrated in Vero 81 cells. The VACV Western Reserve (WR) reference strain (NCBI accession: NC_006998; ATCC: VR-1354) was purchased from ATCC and maintained in RK-13 cells (ATCC: CCL-37) for propagation.

## Study design and sample collection
We conducted a prospective, temporal cohort study to characterize immune responses across multiple populations with different exposures to MPXV and related orthopoxviruses. The study was jointly initiated at the Guangzhou Center for Disease Control and Prevention and Shenzhen Third People's Hospital, with participant enrollment spanning from August 2022 to October 2024. Four participant groups were included in this study: (i) male patients with acute-phase mpox, from whom blood samples were collected at 10 and 20 days after symptom onset at Shenzhen Third People's Hospital to capture early immune responses; (ii) male mpox convalescents in men, recruited and followed temporally at the Guangzhou Center for Disease Control and Prevention for up to 18 months post-onset to assess long-term immune persistence; (iii) age-matched healthy individuals with no known history of orthopoxvirus exposure (VACV-unvaccinated cohort); and (iv) individuals born before 1981 with documented smallpox vaccination history (VACV-vaccinated cohort).

All mpox cases were PCR-confirmed as infections with MPXV clade IIb, consistent with the circulating lineage during the 2022–2024 outbreak in Guangdong. All mpox patients and convalescents presented with moderate clinical manifestations during the course of infection. For convalescent participants, peripheral blood samples were collected at predefined intervals (6, 9, 12, 15, and 18 months post-infection). Peripheral blood mononuclear cells (PBMCs) and plasma were isolated from all participants and stored at −80 °C for subsequent immunological and serological analyses. Baseline demographic and clinical characteristics of all participants are summarized in Table 1. The study was approved by the institutional ethics committees of Guangzhou CDC (GZCDC-ECHR-2023P0059) and Shenzhen Third People's Hospital (2021-030). Written informed consent was obtained from all participants prior to enrollment.

## Detection of MPXV- and VACV-specific binding antibodies by ELISA
Binding antibodies targeting MPXV and VACV membrane-associated proteins were measured by indirect ELISA. High-binding 96-well plates were coated overnight at 4 °C with purified recombinant MPXV and VACV proteins (1 μg/mL), including MPXV A35 (SinoBiological, 40886-V08H), A29 (SinoBiological, 40891-V08E), B6 (SinoBiological, 40902-V08H), E8 (SinoBiological, 40890-V08B), H3 (SinoBiological, 40893-V08H1), and M1 (SinoBiological, 40904-V07H), and VACV A33

(SinoBiological, 40896-V07E), A27 (SinoBiological, 40897-V07E), B5 (SinoBiological, 40900-V08H), H3 (Abmart, EHV4943), D8 (SinoBiological, 40898-V08H), and L1 (SinoBiological, 40903-V07H). After washing with PBST, plates were blocked with 10% skim milk in PBS for 2 h at 37 °C. Four-fold serially diluted human plasma samples (starting at 1:100) were added to the plates and incubated for 2 h at 37 °C. Following further washes, horseradish peroxidase (HRP)-conjugated goat anti-human IgG-Fc antibody (Jackson ImmunoResearch, 109-035-088; 100 μL per well) was applied for 30 min at 37 °C. Color development was achieved using 3,3′,5,5′-tetramethylbenzidine (TMB) substrate (Biohao Biotechnology, N0160), incubated in the dark at room temperature for 10 min, and stopped with 0.2 M sulfuric acid. Optical density (OD) was measured at 450 nm using a microplate reader (BioTek). Endpoint titers were defined as the highest plasma dilution yielding an OD at least 2.1-fold above that of the negative control.

## Preparation of MPXV viral stocks for neutralization assays
Extracellular enveloped virus (EEV) of MPXV was prepared following established protocols[25–27]. Briefly, Vero 81 cells were infected at a multiplicity of infection (MOI) of 0.2 FFU/cell. Culture supernatants were harvested upon complete cytopathic effect (typically observed on day 3 for MPXV clade IIb), clarified by centrifugation at 8000 × g for 15 min, and stored at 4 °C for use within 2 weeks. To eliminate potential contamination with intracellular mature virus (IMV), viral titration was performed in the presence of the IMV-neutralizing monoclonal antibody anti-L1R (clone 7D11). To prepare virus stocks containing both IMV and EEV, Vero E6 cells were infected at an MOI of 0.2 FFU per cell. At 72 h post-infection, both supernatant and cells were collected and subjected to three freeze–thaw cycles to release intracellular viral particles. After clarification by centrifugation at 8000 × g for 15 min, the supernatant was aliquoted and stored at −80 °C for long-term preservation.

## Viral titration of MPXV and VACV
The viral titers of MPXV and VACV were determined by focus-forming assay using Vero 81 and Vero E6 cells, respectively. Serial ten-fold dilutions of viral stocks were inoculated onto confluent monolayers of the corresponding cell lines and incubated at 37 °C with gentle rocking for 1 h. Following removal of the inoculum, cells were overlaid with DMEM containing 2% FBS and incubated at 37 °C in a 5% $CO_2$ atmosphere for 24 h. Cells were then fixed with 4% paraformaldehyde for 30 min at room temperature and permeabilized with 0.2% Triton X-100. Viral foci were detected using a polyclonal anti-vaccinia virus antibody (Abcam, ab35219), followed by horseradish peroxidase (HRP)-conjugated goat anti-rabbit IgG (Jackson ImmunoResearch, 111-035-114). TrueBlue peroxidase substrate (Seracare Life Sciences, 5510−0030) was applied and developed at room temperature for 10 min. Plates were subsequently washed, visualized, and foci were enumerated using an ELISPOT reader (Cellular Technology Ltd).

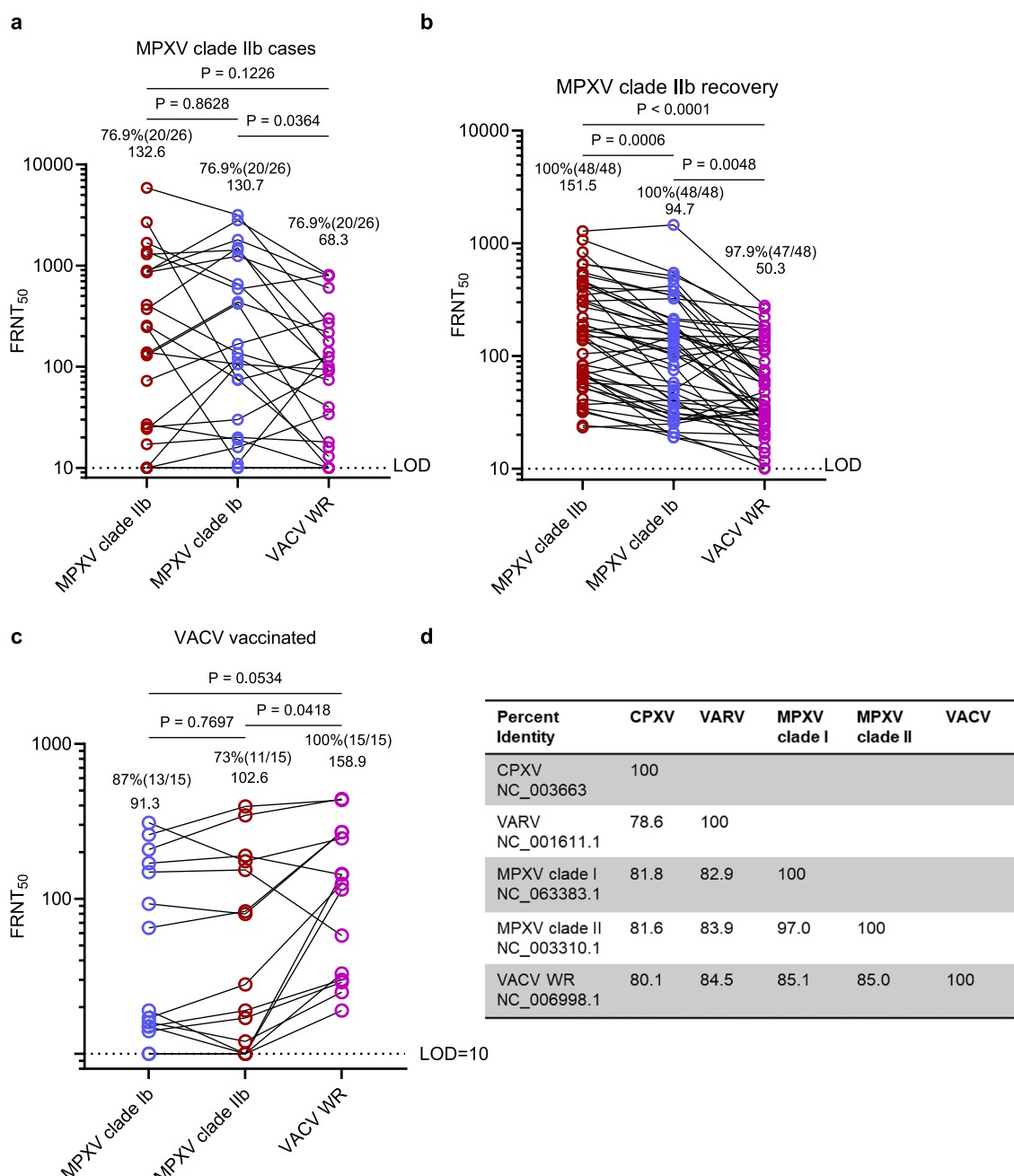

**Fig. 3 | Lineage-specific reductions in neutralization reveal antigenic divergence among orthopoxviruses. a** Neutralizing titer of a subset of acute-phase MPXV samples ($n = 26$, 10–20 days POS) for cross-lineage testing, evaluated against MPXV clade IIb, MPXV clade Ib, and VACV (WR). **b** Neutralizing titers of convalescent plasma (6–18 months POS; $n = 48$) against the same viruses. **c** Neutralizing titers from a subset of historically VACV-vaccinated individuals ($n = 15$) selected for higher titers to allow cross-comparison against the same viruses. Note. Because neutralizing antibody titers were generally too low in historically VACV-immunized individuals (Fig. 2) to allow meaningful comparisons, here we additionally included a subset of historically VACV-immunized individuals with relatively high VACV-neutralizing titers to enable a more reliable assessment of cross-neutralizing activity against different orthopoxviruses. All assays were conducted with 3 technical replicates per sample. **d** Comparative genomic analysis of MPXV clade I, MPXV clade II, and VACV highlighting sequence divergence. The complete genome sequence identities among five representative orthopoxvirus strains, including MPXV clade Ib, IIb, VACV WR, VARV and CPXV. Statistical significance was assessed using one-way ANOVA followed by Tukey's multiple comparisons test. All analyses were conducted using two-sided tests. P-values are displayed as ns for $P > 0.05$, *$P < 0.05$, **$P < 0.01$ and ***$P < 0.001$. All $p$-values were displayed. Dotted lines indicate the limit of detection (LOD = 10) in (**a–c**). Each column presents the mean neutralizing titers and the corresponding seropositivity rates.

## Assessment of MPXV- and VACV-specific neutralizing antibodies by FRNT

Neutralizing antibody responses against MPXV were evaluated using a FRNT, adapted from previously established protocols[25–27]. Briefly, Vero 81 cells were seeded into tissue culture plates 24 h prior to infection. Serial dilutions of serum samples were prepared in DMEM supplemented with 2% heat-inactivated fetal bovine serum in 96-well plates. In parallel, MPXV was diluted to a final concentration of 200 focus-forming units (FFU) per well. Equal volumes of diluted serum and virus were mixed in the presence of 5% guinea pig complement (Bersee, BM361Y) and incubated at 37 °C for 1 h. The plasma-virus mixtures were then added to confluent Vero 81 cell monolayers and incubated for 24 h at 37 °C in 5%

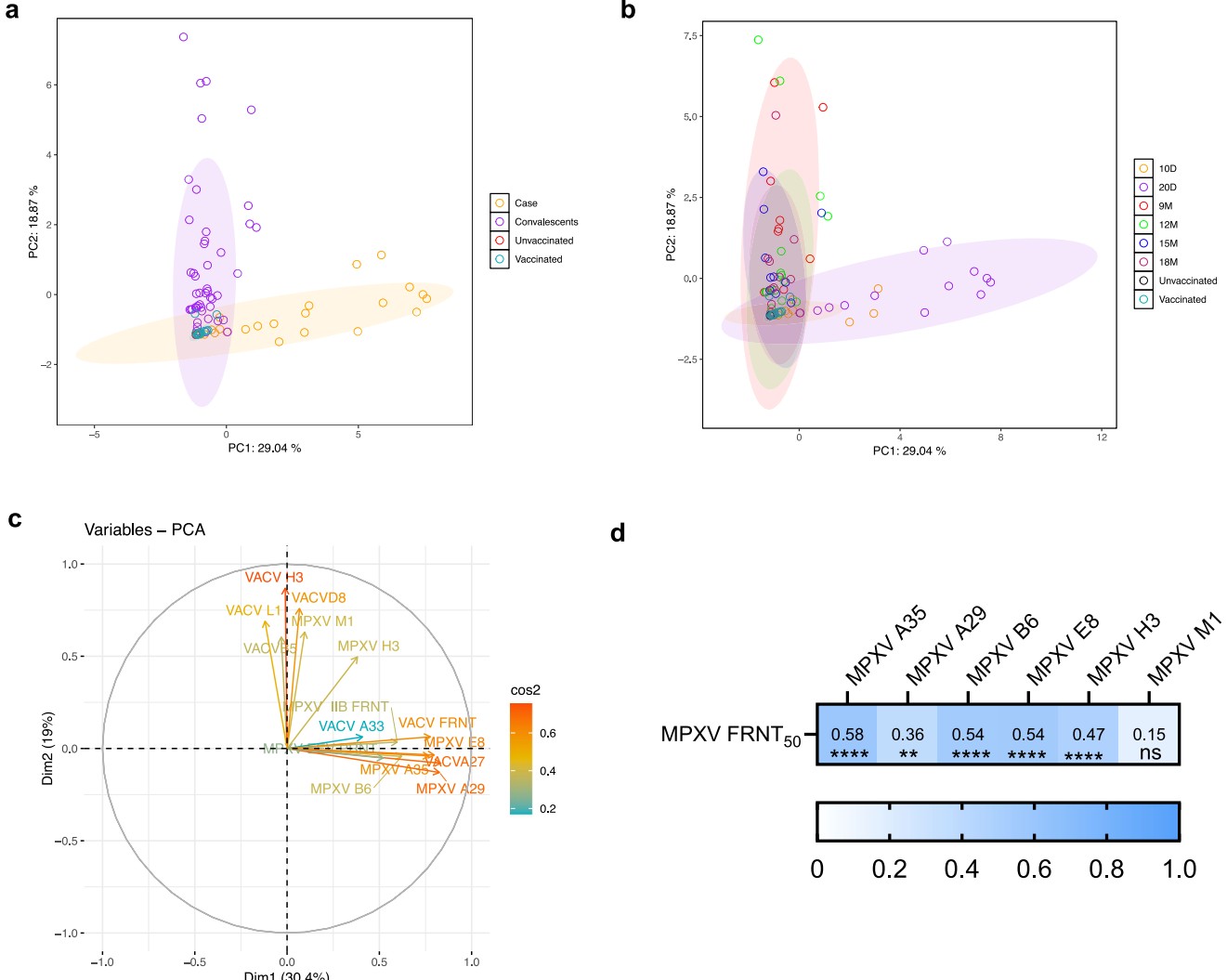

**Fig. 4 | Principal component analysis reveals shared orthopoxvirus antibody responses among MPXV convalescents. a, b** Principal component analysis (PCA) of antibody binding to 6 MPXV, 6 VACV recombinant antigens, and neutralizing antibody against orthopoxvirus, plotted and colored by group. Colored circles represent all samples within that cohort. Four participant groups were included in this study: (1) acute-phase mpox patients, from whom blood samples were collected at 10 days ($n = 13$) and 20 days ($n = 13$) after symptom onset; (2) mpox convalescents who were recruited and followed for up to 18 months post-onset (9 M, $n = 10$; 12 M, $n = 16$; 15 M, $n = 10$; 18 M, $n = 10$); (3) a VACV-unvaccinated cohort ($n = 30$); and (4) a VACV-vaccinated cohort ($n = 30$). **c** Biplot of PCA, highlighting that a number of binding and neutralizing data, which can be used for each particular group. **d** Correlation analysis of MPXV-specific IgG antibody and neutralizing titer among MPXV convalescents was performed using Pearson's correlation coefficient, with all tests conducted as two-sided. Color scale represents the value of the correlation between two variables. $p$-values are displayed as ns for $P > 0.05$, *$P < 0.05$, **$P < 0.01$ and ***$P < 0.001$.

$CO_2$. Subsequent fixation, permeabilization, immunostaining, color development, and foci enumeration were performed as described above. Neutralization data were analyzed using GraphPad Prism (v8.0). Neutralization of VACV was assessed using a microneutralization assay following the same procedure as for MPXV, with minor modifications specific to VACV infection in Vero E6 cells. Because overall neutralizing antibody titers in historically VACV-immunized individuals were relatively low (mean $PRNT_{50} = 57$; Fig. 2b), making direct lineage comparisons unreliable, we additionally selected a subset of participants with higher VACV-neutralizing titers (Fig. 3c). This subset enabled a more robust evaluation of cross-neutralizing activity across orthopoxvirus lineages while reducing the confounding effects of low baseline titers.

**Plaque reduction neutralization test (PRNT) for detection of MPXV EEV-specific neutralizing antibodies**
Neutralizing antibody responses against the EEV form of MPXV were evaluated using a modified plaque reduction neutralization test (PRNT). Vero 81 cells were seeded at a density of $2 \times 10^5$ cells per well in

12-well plates and incubated overnight at 37 °C to form a confluent monolayer. Plasma samples were heat-inactivated and subjected to two-fold serial dilutions in D2 medium (DMEM supplemented with 2% fetal bovine serum). To selectively assess neutralization of the EEV form, diluted plasma samples were mixed at a 1:1 ratio with MPXV in the presence of a saturating concentration of anti-IMV monoclonal antibody 7D11, which effectively neutralizes the IMV population. The mixtures were incubated at 37 °C for 1 h to facilitate antigen–antibody complex formation. After incubation, the plasma–virus mixtures were added to Vero 81 cell monolayers and allowed to adsorb for an additional hour at 37 °C. Cells were then overlaid with DMEM supplemented with 2% fetal bovine serum and 0.8% carboxymethylcellulose to limit viral spread. After 72 h of incubation, the overlay was removed, and the cell monolayers were fixed with 4% paraformaldehyde and stained with 0.1% crystal violet solution to visualize plaques. The EEV-specific neutralization efficacy was determined by calculating the percentage reduction in plaque numbers relative to virus-only controls. Data analysis was performed using GraphPad Prism (v8.0; GraphPad Software).

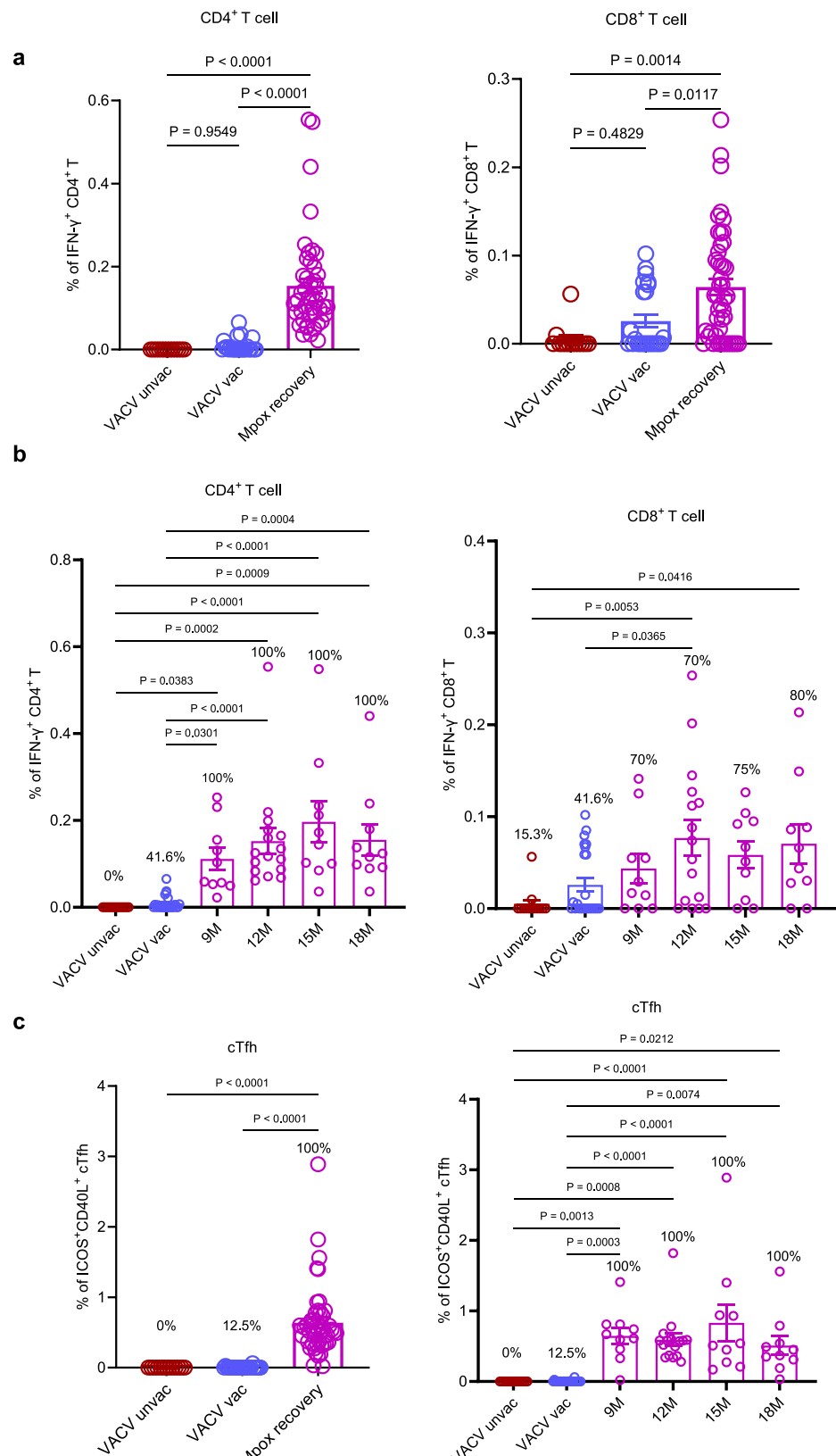

### Assessment of MPXV-specific T cell responses using heat-inactivated viral stimulation

Peripheral blood mononuclear cells (PBMCs) were isolated from MPXV convalescent individuals and seeded at $5 \times 10^5$ to $1 \times 10^6$ cells per well

in 96-well round-bottom plates. To evaluate MPXV-specific T-cell immunity, PBMCs were stimulated with heat-inactivated (56 °C for 30 min) MPXV-infected Vero 81 cell supernatant at a final concentration corresponding to approximately $2 \times 10^5$ FFU per test, or with

**Fig. 5 | Sustained MPXV-specific T cell responses detected in convalescent PBMCs up to 18 months post-onset. a** Comparative analysis of MPXV-specific T cell responses among MPXV convalescent, VACV-vaccinated, and unvaccinated individuals. **b** The frequencies of MPXV-specific CD4[+] and CD8[+] T cells in PBMCs were measured at multiple time points post-recovery. **c** Frequencies of activated cTfh in PBMCs at various time points post-onset are presented. Specific T cell responses shown in (**a**–**c**) are background-subtracted values. All assays were conducted with three technical replicates per sample. Three participant groups were included in this study: mpox convalescents who were recruited and followed for up to 18 months post-onset (9 M, $n = 10$; 12 M, $n = 16$; 15 M, $n = 10$; 18 M, $n = 10$), a VACV-unvaccinated cohort ($n = 13$), and a VACV-vaccinated cohort ($n = 24$). Statistical significance was evaluated using one-way ANOVA followed by Tukey's multiple comparisons test to control for family-wise error rates. All analyses were conducted as two-sided. Error bars represent mean ± SD. *P* values are indicated as follows: ns (not significant) for $P > 0.05$; *$P < 0.05$; **$P < 0.01$; ***$P < 0.001$; ****$P < 0.0001$. Only comparisons with *p*-values ≤ 0.05 are shown.

control non-infected supernatant as negative controls. Following 16–18 h of stimulation, brefeldin A (BioLegend) was added for an additional 6 h to inhibit cytokine secretion. After stimulation, cells were washed and incubated at room temperature with True-Stain Monocyte Blocker™ and Human TruStain FcX™ for 10 min to reduce nonspecific antibody binding. Dead cells were excluded using a viability dye, followed by surface staining with antibodies targeting CD3, CD4, CD8, CD56, CCR7, CXCR5, ICOS and CD45RA at room temperature for 30 min. Cells were then fixed and permeabilized using reagents from BD Biosciences, followed by intracellular cytokine staining for CD40L, IFN-γ, TNF, and IL-2 to profile MPXV-specific T-cell responses. Data acquisition was performed on a BD FACSymphony™ S6, and analyses were conducted using FlowJo software (v10.6.2; BD Biosciences). CD4[+], CD8[+] and cTfh T-cell responses were quantified to characterize the cellular immune response to MPXV in convalescent individuals. The following reagents and fluorochrome-conjugated anti-human monoclonal antibodies were used: True-Stain Monocyte Blocker™ (Biolegend, 426103), Human TruStain FcX™ (Biolegend, 422302), CD3-BUV395 (BD, 563546), CD4-BV510 (BD, 562970), CD8-AF647 (Biolegend, 344726), CD56-BV650 (BD, 564057), CXCR5-PerCP-Cy5.5(BD, 562781), CD45RA-BUV737 (BD, 612846), CCR7-APC-Fire750 (Biolegend, 353246), ICOS-BB515 (BD, 564549), CD40L-PE (Biolegend, 310806), IFN-γ-BV421 (Biolegend, 506538), TNF-α-PE-Cy7 (Biolegend, 502930), IL-2-BV605 (BD, 564165) and Fixable Viability Stain 440UV (BD, 566332). A positive T-cell response was defined as a detectable IFN-γ[+] T-cell frequency upon antigen-specific stimulation, where the background-subtracted value (antigen-stimulated minus unstimulated control) exceeded zero. Representative gating strategies and flow cytometry plots for negative controls are shown in Supplementary Figs. 9 and 10.

### Statistical analysis

Associations between binding, neutralizing antibody and T cell responses were evaluated using Spearman's rank correlation coefficients. All statistical analyses were performed in GraphPad Prism (version 8.0). Neutralizing antibody titers ($FRNT_{50}$) were determined by non-linear regression fitting. Group comparisons were assessed using the Mann–Whitney U test, with significance defined as $p < 0.05$ (*$p < 0.05$; **$p < 0.01$; ***$p < 0.001$; ****$p < 0.0001$). Temporal trends in anti-orthopoxvirus protein IgG levels and neutralizing titers were visualized using scatter plots with locally estimated scatterplot smoothing (LOESS). Blue lines represent LOESS-smoothed trends, and gray shaded areas indicate 95% confidence intervals.

### Reporting summary

Further information on research design is available in the Nature Portfolio Reporting Summary linked to this article.

## Data availability

Genome sequences of the MPXV isolates generated in this study have been deposited in GenBank under accession numbers PX488348 (clade IIb) and PX488349 (clade Ib). Data from the present study are available upon reasonable request from the corresponding author. Patient-related data were included in supplementary table. Source data are provided with this paper.

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

## Acknowledgements

This project was supported by grants from the National Key Research and Development Program of China (2024YFA0920001 to Y.Q.W.), the National Natural Science Foundation of China (82572034 to Y.Q.W., 82025001, 82495200 and 82495203 to J.C.Z.), Guangzhou National Laboratory and State Key Laboratory of Respiratory Disease (GZNL2024B01001 to Y.Q.W.), Guangdong Basic and Applied Research Projects (2023B1515020040 to Y.Q.W.), Science and Technology Planning Project of Guangzhou City (2024A03J1230 to J.C.Z., 2025B04J0006 to L.Z., 2023A04J1259 to L.J.Z), Major Demonstration Project for Scientific and Technological Innovation of Inner Mongolia Autonomous Region (2025ZDSF0021), Key Research and Development and Achievement Transformation Project (2025YFDZ0145), the Key Project of Medicine Discipline of Guangzhou (2025-2027-11) and the Science and Technology Project of General Administration of Customs, P.R.China (2025HK221 to L.Z.). We thank the Biobank for Respiratory Disease in the National Clinical Research Center for Respiratory Disease (BRD-NCRCRD, Guangzhou, Southern China).

## Author contributions

J.C.Z., L.Z., P.Z.Q., Y.Y., and Y.Q.W. conceived and coordinated the project. Y.Q.W., L.Z., R.X.C., J.T.C., and L.J.Z. collected clinical samples and evaluated neutralizing activity in vitro. X.D.X., Q.E.Z., P.L.W., Z.Y.Z., Y.Y.Z., and L.C. conducted ELISA assays to measure MPXV- and VACV-specific binding antibody levels. C.J.C. and A.R.Z. carried out T cell response experiments and performed data analysis. C.J.C., X.X.L., and J.J.G. performed the statistical analyses. S.X.L., X.Y.X., B.Q., S.X.H., Z.W.L., and H.S.B. prepared viral stocks and optimized the detection assays. Q.T.H., J.X.Z., and Y.X.S. provided technical support. Y.Q.W. drafted and revised the manuscript. All authors reviewed and approved the final version.

## Competing interests

The authors declare no competing interests.
