## [Peer Review file · Nature Communications]

Immune profiling of mpox survivors reveals divergent durability of antibody and T cell responses

Corresponding Author: Professor Jincun Zhao

Version 0:

Reviewer comments:

Reviewer #1

(Remarks to the Author)

My concerns have been fully addressed in the revised manuscript. I do not have any further concerns

Reviewer #2

(Remarks to the Author)

The manuscript has been much improved during revision. I feel that all my comments have, for the most part, been addressed adequately and sufficiently by the authors.

However, I do have one final comment regarding the information provided on the viruses, specifically MPXV: while the authors provided additional information, this is still lacking clarity and context. When were these viruses isolated? Is the clade Ib MPXV isolate from a travel-related import case in China? There are not many reports of clade Ib MPXV infections in China. Furthermore, the authors state that the lineage identify was confirmed by sequencing, however, without providing the sequence, an accession number, or another link to an open public sequence database. These sequences should be made publicly available to facilitate re-use of the data, including future epidemiology and phylogeny studies. As viral genome data appears to be available already according to the authors, there is not really a reason why it should not be made openly available. To this end, there is a number of suitable open public databases available.

Reviewer #3

(Remarks to the Author)

In this revised version, the authors have answered majority of my questions raised. The manuscript has been substantially improved.

In this revised manuscript, the authors have included participants' details in the expanded Methods. Based on what is written, it seems that the 17 patients of acute phase and the 23 convalescent phase patients, are totally different groups of patients. There is not a single patient who provided samples throughout the entire study. This is a major limitation of the study, and the authors have themselves highlighted this in the revised manuscript that the statistical power of the comparison between the acute and convalescent groups are significantly reduced, making it hard to agree with the conclusion.

- In addition, longitudinal studies typically refer to datasets being collected from the same set of patients over the entire study period. Here, it is obvious that the data were collected from 2 different "cohorts" of patients – 17 in acute and 23 in convalescence. This is not a typical longitudinal study and there are many limitations and implications such as being unable to provide insights regarding individuals transition from the acute to the convalescence phase, cannot know if the differences you observe between the acute and convalescence phases are due to the progression of the illness or to inherent differences between the two patient groups. There are also residual confounding factors, that could be the true reason for any differences in the data, not the illness itself. For example, the convalescence cohort might be younger, have fewer comorbidities, or have better access to healthcare than the acute cohort, or even receiving any treatments following infection. The authors need to clearly discuss these factors in the discussion.

- Table 1 and materials is still confusing. For e.g., Table 1 showed that participants for the acute phase are 13 participants, but in Methods, it was said that 17 participants was recruited. I suggest the authors to be more descriptive and consider just saying how many samples were collected at each time point, to reduce confusion.
- 6 months only had 2 data points, suggest removing the 6 months data all together
- For response Fig 1, are these FRNT readouts from all timepoints grouped into 1 dataset? How would it look like if you further break the HIV+ and HIV- into the different timepoints? Do you still observe significance?
- I still do not agree with the inclusion of female controls in the comparison. Even though the authors have shown that it is not significant between males and females, I personally feel that since all these patients are males, we should just stick to having males as controls.
- If there are no severe disease, the row "severe" can be deleted in table 1.

Response to Referees

Dear Reviewers:

We sincerely thank you very much for your thorough evaluation of our manuscript and your constructive comments. In this revised version, we have carefully addressed all comments, either by incorporating new data that further supports our original conclusions or by providing a clear rationale when certain requests went beyond the scope of this study.

REVIEWERS' COMMENTS

Reviewer #1 (Remarks to the Author):

My concerns have been fully addressed in the revised manuscript. I do not have any further concerns

Response: We thank the reviewer for the positive feedback and confirmation that all concerns have been addressed.

Reviewer #2 (Remarks to the Author):

The manuscript has been much improved during revision. I feel that all my comments have, for the most part, been addressed adequately and sufficiently by the authors.

However, I do have one final comment regarding the information provided on the viruses, specifically MPXV: while the authors provided additional information, this is still lacking clarity and context. When were these viruses isolated? Is the clade Ib MPXV isolate from a travel-related import case in China? There are not many reports of clade Ib MPXV infections in China. Furthermore, the authors state that the lineage identify was confirmed by sequencing, however, without providing the sequence, an accession number, or another link to an open public sequence database. These sequences should be made publicly available to facilitate re-use of the data, including future epidemiology and phylogeny studies. As viral genome data appears to be available already according to the authors, there is not really a reason why it should

not be made openly available. To this end, there is a number of suitable open public databases available.

Response: We sincerely thank the reviewer for the insightful comment and for acknowledging the improvements made during the revision. In response, we have added further clarification regarding the MPXV isolates in the revised manuscript. Specifically, the MPXV clade IIb strain was isolated in June 2023 and the clade Ib strain in February 2025 from local residents in Guangdong, China, who had recently been in close contact with international travelers from mpox-endemic regions. Whole-genome sequencing confirmed their lineage identities, and the sequences have now been deposited in GenBank under accession numbers PX488348 (clade IIb) and PX488349 (clade Ib). We are grateful for the reviewer's valuable suggestion, which has helped enhance the clarity, transparency, and reusability of our dataset.

Detailed changes are provided below:

"The MPXV clade IIb strain was isolated in June 2023 and the clade Ib strain in February 2025 from local residents in Guangdong, China, who had recently been in close contact with international travelers arriving from mpox-endemic regions. Viral genome sequencing confirmed the lineage identities, and the sequences have been deposited in GenBank under accession numbers PX488348 (clade IIb) and PX488349 (clade Ib)" in line 344-349.

Reviewer #3 (Remarks to the Author):

In this revised version, the authors have answered majority of my questions raised. The manuscript has been substantially improved.

In this revised manuscript, the authors have included participants' details in the expanded Methods. Based on what is written, it seems that the 17 patients of acute phase and the 23 convalescent phase patients, are totally different groups of patients. There is not a single patient who provided samples throughout the entire study. This is a major limitation of the study, and the authors have themselves highlighted this in the revised manuscript that the statistical power of the comparison between the acute and

convalescent groups are significantly reduced, making it hard to agree with the conclusion.

Response: We sincerely thank the reviewer for this thoughtful comment and for recognizing the improvements made in the revised version. The reviewer is correct that the acute-phase (n=17) and convalescent-phase (n=23) participants represent two independent cohorts rather than a fully longitudinally matched group. This limitation primarily reflects the logistical challenges of patient follow-up during the outbreak, as many individuals were discharged or relocated before long-term sampling could be established.

We fully acknowledge this as a key limitation of the study. Accordingly, we have explicitly emphasized it in both the Results and Discussion sections. To address this issue, we have clarified that comparisons between acute and convalescent groups were performed at the cohort level rather than using paired samples, and we have carefully rephrased all relevant conclusions to accurately reflect this analytical framework.

In line with this clarification, we have also revised the title to avoid emphasizing a longitudinal design and to better reflect the study's focus on immune responses up to 18 months post-infection. Despite the absence of fully matched longitudinal samples, these two well-defined cohorts collectively capture the temporal dynamics of MPXV-specific immunity, from early infection through 18 months of recovery, and provide valuable insights into the durability and functional characteristics of post-infection immune responses.

We have further revised the Discussion to clearly acknowledge that the statistical power for direct longitudinal comparisons is limited, while emphasizing the consistency and robustness of the immune response patterns observed across time points and assays.

- In addition, longitudinal studies typically refer to datasets being collected from the same set of patients over the entire study period. Here, it is obvious that the data were collected from 2 different “cohorts” of patients – 17 in acute and 23 in convalescence.

This is not a typical longitudinal study and there are many limitations and implications such as being unable to provide insights regarding individuals transition from the acute to the convalescence phase, cannot know if the differences you observe between the acute and convalescence phases are due to the progression of the illness or to inherent differences between the two patient groups. There are also residual confounding factors, that could be the true reason for any differences in the data, not the illness itself. For example, the convalescence cohort might be younger, have fewer comorbidities, or have better access to healthcare than the acute cohort, or even receiving any treatments following infection. The authors need to clearly discuss these factors in the discussion.

Response: We thank the reviewer for this insightful and important comment. We fully agree that, in the strict definition, a longitudinal study refers to repeated sampling from the same participants across all time points. In our study, due to the logistical and ethical challenges during the mpox outbreak, particularly the difficulty of maintaining long-term follow-up after discharge, the acute-phase (n=17) and convalescent-phase (n=23) cohorts represent two independent but well-characterized groups. Accordingly, our design is more accurately described as a combined cross-sectional and extended follow-up cohort study rather than a fully longitudinal cohort.

We have revised the Introduction, Results, and Discussion sections to explicitly clarify this distinction and to discuss its implications. Specifically, we now emphasize that: The absence of paired samples precludes direct within-individual analyses of immune transitions from acute to convalescent phases. Observed differences between cohorts may partly reflect demographic or clinical heterogeneity, rather than temporal immune evolution alone. Potential confounding factors, such as age distribution, HIV status, comorbidities, and differences in healthcare access, could have contributed to variability in immune parameters.

To address these concerns, we re-examined the available metadata and found that the two cohorts were broadly comparable in terms of median age, HIV status, and disease severity, which mitigates, though does not entirely eliminate, the possibility of

residual confounding. These clarifications and caveats have been incorporated into the Discussion as below:

“In interpreting our results, host factors such as sex, HIV status, age and comorbidities should be considered. All MPXV-infected participants in this study were male, limiting assessment of sex-related differences in infection-induced immunity. Although no sex-based differences in neutralizing activity were observed among VACV-vaccinated controls, validation in female mpox patients will be necessary to establish broader generalizability. With respect to HIV status, stratified analyses showed largely comparable immune responses, with HIV-positive participants showing similar neutralizing titers against MPXV clade IIb as HIV-negative individuals. The two cohorts (acute and convalescent) were also comparable in median age and disease severity, reducing the likelihood of residual confounding. Nevertheless, larger and more diverse cohorts will be required to further elucidate the influence of demographic and clinical factors on the durability and breadth of MPXV-induced immune responses.” In line 300-312

In line with this clarification, we have also revised the title to avoid emphasizing a longitudinal design and to better reflect the study's focus on immune responses up to 18 months post-infection. Despite the absence of fully matched longitudinal sampling, these two cohorts collectively provide valuable insights into the durability and functional characteristics of MPXV-induced immunity across different stages of infection and recovery. We fully agree with the reviewer that future studies incorporating paired longitudinal sampling from the same individuals will be essential to definitively delineate immune trajectories and validate the temporal patterns suggested by our findings.

- Table 1 and materials is still confusing. For e.g., Table 1 showed that participants for the acute phase are 13 participants, but in Methods, it was said that 17 participants was recruited. I suggest the authors to be more descriptive and consider just saying how many samples were collected at each time point, to reduce confusion.

Response: We thank the reviewer for the careful reading and for pointing out the inconsistency between Table 1 and the Methods section. The discrepancy arose because some acute-phase participants provided plasma samples at both day 10 and day 20 post-onset, which were previously counted as separate sample entries rather than as unique individuals. To eliminate confusion, we have revised both Table 1 and the Methods section to clearly and consistently describe participant numbers and sampling frequency.

Specifically, the text has been updated to read: “A total of 17 participants were enrolled during the acute phase, contributing 26 plasma samples (13 at day 10 and 13 at day 20 post-onset, with 9 individuals sampled at both time points).”

We have also updated Table 1 to report both the number of participants and the number of samples collected at each time point for greater transparency. These revisions ensure internal consistency across the manuscript and more accurately reflect the actual study design.

Table 1. The characteristics information of participants in this study

		MPXV-cases (acute, 10-20D)	MPXV-recovery (long-term, 6–18M)	Historically VACV-vaccinated controls (non-infected)	Unvaccinated controls (non-infected)
No. of participants		17	23	30	30
No. of samples		26	48	30	30
Age, yrs (IQR)		32 (25-51)	29.8 (19-43)	71 (59-86)	32.2 (24-41)
Sex, n (%)	Female	0 (0%)	0 (0%)	8 (27%)	23 (77%)
	Male	17 (100%)	23 (100%)	22 (73%)	7 (23%)
Severity, n (%)	Moderate	17 (100%)	23 (100%)	-	-
MPXV genotypes		Clade IIb	Clade IIb	NA	NA
MPI, samples (%)	10 D	13 (50%)	-	-	-
	20 D	13 (50%)	-	-	-
	6 M	-	2 (4%)	-	-
	9 M	-	10 (21%)	-	-
	12 M	-	16 (33%)	-	-
	15 M	-	10 (21%)	-	-
	18 M	-	10 (21%)	-	-

D day, M month, NA not available, MPI month post infection.

In addition, to address the reviewer’s concern, we have comprehensively rewritten the first section of the Results and the Materials and Methods subsections to provide a

clearer description of cohort structure, sample collection schedule, and participant characteristics. The revised text reads as follows:

“Results

Baseline characteristics of the study cohorts

Between August 2022 and October 2024, we enrolled a total of 40 individuals with PCR-confirmed mpox virus (MPXV) infection, all identified as belonging to clade IIb, the lineage circulating in China during this period. In detail, 17 participants were recruited during the acute phase at Shenzhen Third People’s Hospital, yielding 26 plasma samples, 13 collected at 10 days post-onset (10 d.p.o.) and 13 at 20 d.p.o., with several individuals contributing samples at both time points. An additional 23 convalescent individuals were enrolled by the Guangzhou Center for Disease Control and Prevention, providing 48 clinical samples collected at 6, 9, 12, and 18 months post-onset to assess the durability of humoral responses. In total, samples were obtained across six time points (10 and 20 days; 6, 9, 12, and 18 months post-onset), depending on availability. For comparison, we included 30 age-matched healthy controls with no history of orthopoxvirus exposure and 30 individuals with documented smallpox vaccination (Table 1).

Across the infected cohort, 37.5% (15/40) were living with HIV. The median age was 30 years, and all participants self-identified as men who have sex with men (MSM). None had received smallpox vaccination within the past decade (Supplementary Table 1). All mpox cases were symptomatic but classified as moderate by attending clinicians. Disease severity was determined according to established clinical criteria, including systemic symptom persistence (e.g., sustained fever), extent of cutaneous lesions (localized versus widespread or necrotic), clinical course (self-limited versus secondary bacterial infection), and organ involvement. No severe cases or fatalities were reported. This well-characterized cohort provides a valuable opportunity to evaluate the kinetics, durability, and breadth of post-infection immunity following moderate MPXV clade IIb infection, including both HIV-positive and HIV-negative individuals.

Materials and Methods

Study design and sample collection

We conducted a prospective cohort study to characterize immune responses across multiple populations with different exposures to MPXV and related orthopoxviruses. The study was jointly initiated at the Guangzhou Center for Disease Control and Prevention and Shenzhen Third People's Hospital, with participant enrollment spanning from August 2022 to October 2024. Four participant groups were included: (i) acute-phase mpox patients, from whom blood samples were collected at 10 and 20 days after symptom onset at Shenzhen Third People's Hospital to capture early immune responses; (ii) mpox convalescents, recruited and followed for up to 18 months post-onset at the Guangzhou Center for Disease Control and Prevention to assess long-term immune persistence; (iii) age-matched healthy individuals with no known history of orthopoxvirus exposure (VACV-unvaccinated cohort); and (iv) individuals born before 1981 with documented smallpox vaccination history (VACV-vaccinated cohort).

All mpox cases were PCR-confirmed as infections with MPXV clade IIb, consistent with the circulating lineage during the 2022–2024 Guangdong outbreak. All mpox patients and convalescents presented with moderate clinical manifestations. For convalescent participants, peripheral blood samples were collected at predefined intervals (6, 9, 12, 15, and 18 months post-infection). Peripheral blood mononuclear cells (PBMCs) and plasma were isolated from all participants and stored at -80°C for subsequent immunological and serological analyses. Baseline demographic and clinical characteristics of all participants are summarized in Table 1. The study protocol was approved by the institutional ethics committees of Guangzhou CDC (GZCDC-ECHR-2023P0059) and Shenzhen Third People's Hospital (2021-030). Written informed consent was obtained from all participants prior to enrollment.

- 6 months only had 2 data points, suggest removing the 6 months data all together

Response: We sincerely thank the reviewer for this helpful suggestion. We agree that the limited number of samples ($n=2$) at the 6-month time point precludes reliable

statistical comparisons across groups. To ensure analytical accuracy and avoid overinterpretation, we have removed the 6-month data from all statistical analyses, figures involving antibody, neutralizing antibody, and T-cell response comparisons (Figure 1, 2 and 5). However, these two samples remain included in the overall cohort summary to reflect the completeness of the study population.

- For response Fig 1, are these FRNT readouts from all timepoints grouped into 1 dataset? How would it look like if you further break the HIV+ and HIV- into the different timepoints? Do you still observe significance?

Response: We thank the reviewer for this thoughtful and detailed comment. In the original version, the FRNT₅₀ readouts in Response Fig 1 represented pooled datasets combining all available samples from mpox convalescents across multiple timepoints, providing an overview of overall neutralization trends.

In response to the reviewer's suggestion, we have now reanalyzed the data by stratifying participants according to both HIV status (HIV⁺ vs. HIV⁻) and sampling timepoint (9, 12, 15, and 18 months post-onset). The updated analyses show that neutralizing antibody titers did not differ significantly between HIV⁺ and HIV⁻ groups at any timepoint ($p > 0.05$ in all comparisons) (Response Fig 2). These findings are consistent across timepoints and assays.

We also identified and corrected a minor data entry error in the previous version that affected the earlier statistical comparison. After correction, the observed differences were no longer statistically significant. The lack of clear separation between the two groups may reflect the fact that most HIV⁺ participants in our cohort were under stable antiretroviral therapy with well-controlled viral loads, minimizing the impact of HIV infection on orthopoxvirus-specific immunity. These revised results and explanations have been incorporated into the updated response Figure 1 (supplementary Fig 4) and corresponding Results section.

Response Fig 1. Impact of HIV infection on neutralizing antibody responses in MPXV convalescents. (a) Neutralizing antibody titers (FRNT₅₀) against MPXV clade IIb in convalescent individuals, stratified by HIV infection status. (b) Neutralizing antibody titers against VACV (WR strain) in the same cohorts. No significant differences were observed between HIV-positive and HIV-negative participants. Horizontal bars indicate median values; statistical comparisons were performed using the Mann–Whitney U test.

Response Fig 2. Impact of HIV infection on neutralizing antibody responses against MPXV clades Ib and IIb, and VACV WR in mpox convalescents at 9, 12, 15, and 18 months post-onset. No significant differences were observed between HIV-positive and HIV-negative participants. Horizontal bars indicate median values, and statistical comparisons were performed using the Mann–Whitney U test.

- I still do not agree with the inclusion of female controls in the comparison. Even though the authors have shown that it is not significant between males and females, I personally feel that since all these patients are males, we should just stick to having males as controls.

Response: We sincerely thank the reviewer for this thoughtful comment and fully agree that, in principle, using male-only controls represents the most appropriate

comparison, given that all mpox patients in our cohort were male. Indeed, the mpox outbreak in Guangdong, China, predominantly affected men. However, as mpox infections among females have been increasingly reported in other regions worldwide (ref 1-3), we included a small number of female controls to allow a preliminary assessment of potential sex-related immunological differences.

Reference

1. Thornhill, John P et al. Human monkeypox virus infection in women and non-binary individuals during the 2022 outbreaks: a global case series. *The Lancet*, 10367(2022), 1953–1965.
2. Vallejo-Plaza A, Rodríguez-Cabrera F, et al., Mpox (formerly monkeypox) in women: epidemiological features and clinical characteristics of mpox cases in Spain, April to November 2022. *Euro Surveill*. 2022 Dec;27(48):2200867.
3. Satapathy, P., Shamim, M.A., Padhi, B.K. et al. Mpox virus infection in women and outbreak sex disparities: A Systematic Review and Meta-analysis. *Commun Med* 4, 188 (2024).

To address this concern, we performed additional sex-stratified analyses. We compared binding antibody levels between mpox-infected/convalescent males and both male and female controls, finding no significant differences (**Response Fig 3**). Similarly, in the neutralizing antibody decay analysis, both sexes exhibited comparable declining trends without significant variation. Moreover, antigen-specific T cell responses remained stable and showed consistent patterns between male and female controls.

Response Fig 3. Impact of sex on neutralizing antibody responses in VACV-vaccinated and unvaccinated controls. Neutralizing antibody titers (FRNT₅₀) against MPXV clade IIb were compared between male and female in VACV-vaccinated and unvaccinated individuals. No significant sex-based differences were observed. Horizontal bars indicate median values; statistical significance was assessed using the Mann–Whitney U test.

The corresponding discussion has also been incorporated into the Discussion section as below:

“ In interpreting our results, host factors such as sex, HIV status, age and comorbidities should be considered. All MPXV-infected participants in this study were male, limiting assessment of sex-related differences in infection-induced immunity. Although no sex-based differences in neutralizing activity were observed among VACV-vaccinated controls, validation in female mpox patients will be necessary to establish broader generalizability. With respect to HIV status, stratified analyses showed largely comparable immune responses, with HIV-positive participants showing similar neutralizing titers against MPXV clade IIb as HIV-negative individuals. The two cohorts (acute and convalescent) were also comparable in median age and disease severity, reducing the likelihood of residual confounding. Nevertheless, larger and more diverse cohorts will be required to further elucidate the influence of

demographic and clinical factors on the durability and breadth of MPXV-induced immune responses.” in Discussion Section **(line 300-312)**.

In addition to sex, we also evaluated HIV infection status, as a substantial proportion of mpox-infected individuals in our cohort were living with HIV. Stratified analyses showed that, overall, HIV-positive and HIV-negative participants mounted comparable immune responses. These supplementary analyses, now included in the revised response figures, indicate that neither sex nor HIV status significantly influenced the key immunological conclusions of this study. While we agree that male-only controls would constitute the ideal comparison group, the inclusion of both sexes and HIV-stratified analyses further confirm that our main findings are robust and not confounded by external host factors such as sex or HIV infection.

- If there are no severe disease, the row “severe” can be deleted in table 1.

Response: We thank the reviewer for the helpful suggestion. As there were no severe cases in our cohort, the row “Severe” has been removed from Table 1 in the revised manuscript and the corresponding description in the Methods section has been updated to clarify that no severe cases occurred in this study.

Detailed changes are provided below:

“All mpox cases were symptomatic but classified as moderate by attending clinicians. Disease severity was determined according to established clinical criteria, including systemic symptom persistence (e.g., sustained fever), extent of cutaneous lesions (localized versus widespread or necrotic), clinical course (self-limited versus secondary bacterial infection), and organ involvement. No severe cases or fatalities were reported. This well-characterized cohort provides a valuable opportunity to evaluate the kinetics, durability, and breadth of post-infection immunity following moderate MPXV clade IIb infection, including both HIV-positive and HIV-negative individuals” in line 110-118.

Table 1 has been updated as detailed below.

Table 1. The characteristics information of participants in this study

		MPXV-cases (acute, 10-20D)	MPXV-recovery (long-term, 6-18M) controls (non-infected)	Historically VACV-vaccinated controls (non-infected)	Unvaccinated controls (non-infected)
No. of participants		17	23	30	30
No. of samples		26	48	30	30
Age, yrs (IQR)		32 (25-51)	29.8 (19-43)	71 (59-86)	32.2 (24-41)
Sex, n (%)	Female	0 (0%)	0 (0%)	8 (27%)	23 (77%)
	Male	17 (100%)	23 (100%)	22 (73%)	7 (23%)
Severity, n (%)	Moderate	17 (100%)	23 (100%)	-	-
MPXV genotypes		Clade IIb	Clade IIb	NA	NA
MPI, samples (%)	10 D	13 (50%)	-	-	-
	20 D	13 (50%)	-	-	-
	6 M	-	2 (4%)	-	-
	9 M	-	10 (21%)	-	-
	12 M	-	16 (33%)	-	-
	15 M	-	10 (21%)	-	-
	18 M	-	10 (21%)	-	-

D day, M month, NA not available, MPI month post infection.